# Genetic Diagnosis for 64 Patients with Inherited Retinal Disease

**DOI:** 10.3390/genes14010074

**Published:** 2022-12-26

**Authors:** Jacob Lynn, Austin Raney, Nathaniel Britton, Josh Ramoin, Ryan W. Yang, Bojana Radojevic, Cynthia K. McClard, Ronald Kingsley, Razek Georges Coussa, Lea D. Bennett

**Affiliations:** 1Department of Pathology, University of Oklahoma Health Sciences Center, Oklahoma City, OK 73104, USA; 2College of Medicine, University of Oklahoma Health Sciences Center, Oklahoma City, OK 73104, USA; 3College of Osteopathic Medicine, Oklahoma State University, Stillwater, OK 74078, USA; 4Department of Ophthalmology, University of Oklahoma Health Sciences Center, Dean McGee Eye Institute, Oklahoma City, OK 73104, USA

**Keywords:** inherited retinal disease, genetic testing, clinical

## Abstract

The overlapping genetic and clinical spectrum in inherited retinal degeneration (IRD) creates challenges for accurate diagnoses. The goal of this work was to determine the genetic diagnosis and clinical features for patients diagnosed with an IRD. After signing informed consent, peripheral blood or saliva was collected from 64 patients diagnosed with an IRD. Genetic testing was performed on each patient in a Clinical Laboratory Improvement Amendments of 1988 (CLIA) certified laboratory. Mutations were verified with Sanger sequencing and segregation analysis when possible. Visual acuity was measured with a traditional Snellen chart and converted to a logarithm of minimal angle of resolution (logMAR). Fundus images of dilated eyes were acquired with the Optos^®^ camera (Dunfermline, UK). Horizontal line scans were obtained with spectral-domain optical coherence tomography (SDOCT; Spectralis, Heidelberg, Germany). Genetic testing combined with segregation analysis resolved molecular and clinical diagnoses for 75% of patients. Ten novel mutations were found and unique genotype phenotype associations were made for the genes *RP2* and *CEP83*. Collective knowledge is thereby expanded of the genetic basis and phenotypic correlation in IRD.

## 1. Introduction

Non-correctable vision impairments are expected to increase to 13 million people by 2050 [1]. One of the leading causes of these impairments is inherited retinal diseases (IRDs) which has a prevalence of 1 in 2000–3000 [2,3]. IRDs encompass a clinically and genetically heterogeneous group of disorders including retinitis pigmentosa (RP), cone-rod dystrophy (CRD), and various forms of macular dystrophy (MD) such as Stargardt disease (STGD1) and pattern dystrophy. This group of inherited disorders involve progressive degeneration of the retina that leads to severe visual impairment and blindness. The most common IRD, RP is characterized by retinal pigment epithelium (RPE) abnormalities and primary loss of rod photoreceptor cells followed by secondary loss of cone photoreceptor cells. Patients typically experience a decline to total loss of night vision during adolescence, followed by decreased peripheral vision in young adulthood, and deterioration of the central vision in later life [4]. Dystrophies initially affecting the macula (e.g., MD, STDG1) are distinct from pan-retinal CRD but often have multiple, similar symptoms that can include uncorrectable visual acuity, decreased color perception (dyschromatopsia), and increased sensitivity to light (photophobia).

The overlapping genetic and clinical spectrum in IRD creates challenges for accurate diagnoses. Technological advances in sequencing strategies allow identification of gene mutations for 60–80% of the patients diagnosed with an IRD [5,6,7,8,9,10,11]. IRDs are predominantly Mendelian and monogenic but continue to confound diagnosis due to incomplete penetrance and variable expression. These diseases are typically classified by inheritance type, age of onset, molecular defect, and by distribution of retinal involvement or fundus appearance. A direct link from the molecular and biochemical variations to clinical manifestations contributes to the ideal, unified subclassification system sought by researchers and clinicians alike. This work contributes to realizing this goal.

Genetic diagnosis is a critical step in identifying patients who could be eligible for FDA-approved gene therapy (RPE65-associated IRD-Luxturna [12,13,14]) or for one of the many gene targeted clinical trials (e.g., *MERTK*, *RS1*, *RPGR*-associated IRDs [15,16,17]). An early and accurate diagnosis enables patients to obtain appropriate visual, social, and psychological support that will guide life planning and mitigate disease effects on lifestyle choices. To address these challenges, the objective of this work was to determine the molecular (genetic) basis of IRD for 64 patients paired with clinical findings to advance our understanding of genotype-phenotype associations for IRD.

## 2. Materials and Methods

Patients were evaluated at Dean McGee Eye Institute (DMEI) at the University of Oklahoma Health Sciences Center (OUHSC) in Oklahoma City, OK. Retrospective data was collected from patients who were seen from 2018 to 2021. All procedures adhered to the Declaration of Helsinki and were approved by institutional ethics review boards. Exclusion criteria included confounding diagnoses such as autoimmune retinopathy, age-related macular degeneration, and history of retinotoxic medications (e.g., pentosan polysulfate or hydroxychloroquine). Demographics in Appendix A include identification (ID) number, gender, race or ethnicity, age at diagnosis, age at genetic testing, diagnosis, and inheritance. ID numbers with a letter after the ID number are family members (relation in parenthesis). The ethnicity of the patients in this cohort were White or Caucasian (WC, *n* = 45, 70.3%), Black or African American (BA, *n* = 3, 4.7%), Asian (*n* = 1, 1.6%), Native American (NA, *n* = 10, 15.6%), unknown or undisclosed (UK, *n* = 1, 1.6%), and bi-racial (*n* = 4, 6.3%).

Genetic testing was performed after obtaining informed, written consent from all patients and family members. Peripheral blood or saliva samples of affected and unaffected family members were collected and targeted next generation sequencing (NGS) was performed externally by CLIA certified laboratories (BluePrint Genetics, Invitae, Molecular Vision Laboratory, or Molecular Genetics Laboratory at OUHSC). This certification obliges them to adhere to federal standards. Their testing panels included 351, 330, 1024, and 251 genes, respectively. Molecular Vision Laboratory’s panel included genes for diseases outside the boundaries of IRD. Current IRD gene panels used for NGS are publicly available at Blueprint Genetics, Invitae, and Molecular Vision Laboratory.

Testing performance was provided on individual patient reports. Sequencing was performed using hybridization-based target capture protocols. To summarize the testing information for the aforementioned companies, sequences were mapped to the human reference genome GRCh37/hg19. Genes were excluded from reporting when NGS coverage was suboptimal, i.e., “>90% of the gene’s target nucleotides were not covered at >20x with a mapping quality score of MQ > 20 reads.” Deletions and duplications were identified with proprietary bioinformatics pipelines developed by the individual companies. When deletions, duplications, and pathogenic variants did not meet NGS quality metrics of a given laboratory, orthogonal assays were performed to confirm the results. These assays included, but were not limited to, Pacific Biosciences SMRT sequencing, MLPA, MLPA-seq, Array CGH, and RT-PCR. Each lab designed their own primers and do not provide them on the patient reports. Analytical sensitivity exceeded 99% for variants, including indels less than 15 bp in length for all laboratories. Variants were classified according to the American College of Medical Genetics and Genomics (ACMG) guidelines. Pathogenesis of identified variants was determined by the consequences of the change (e.g., amino acid substitution), evolutionary conservation, the number of reference population databases, and mutation databases containing the recorded variant. These included but were not limited to gnomAD, ClinVar, HGMD Professional, and Alamut Visual. Variant calling, methods, and data analysis (or contact information for requests) for individual laboratories can be found at https://blueprintgenetics.com/data-analysis/, https://www.invitae.com/static/data/WhitePaper_Variant-Classification-Method.pdf, https://www.molecularvisionlab.com/company/publications/, or https://genetics.ouhsc.edu/ (accessed on 23 December 2022).

Visual acuity was measured with a traditional Snellen chart in clinic at DMEI and converted to a logarithm of minimal angle of resolution (logMAR). Fundus images of dilated eyes were acquired with the Optos^®^ camera (Optos PLC, Dunfermline, UK). Horizontal line scans were obtained with spectral-domain optical coherence tomography (SDOCT; Spectralis HRA; Heidelberg Engineering, Heidelberg, Germany).

## 3. Results

### 3.1. Clinical Assessment

Genetic testing was performed on 64 patients (*n* = 29 male, 45%; *n* = 35 female, 55%) diagnosed with an IRD from 53 different families. Clinical diagnosis occurred at an average age of 31.4 ± 15.3 years whereas genetic testing was performed when patients were at the average age of 47.7 ± 16.6 years (Appendix A). Clinical diagnoses included RP (*n* = 32, 50.0%), MD (*n* = 24, 37.5%), and CRD (*n* = 8, 12.5%). Refraction and BCVA are shown in Appendix A. Ophthalmic evaluations were conducted on 60 patients with IRD (29 RP, 7 CRD, and 23 MD). Overall, patients with RP and CRD had the broadest range of clinical features including pallor of the optic disc, arteriolar narrowing, bone spicules, peripapillary atrophy (PPA), cystoid macular edema (CME), RPE changes in the macula, vitreous syneresis, posterior vitreous detachment (PVD), and/or epiretinal membrane (ERM; Figure 1A). Clinical examination showed typical clinical features (pallor of the optic disc, arteriolar narrowing, and bone spicules) for 17 of 29 individuals diagnosed with RP. Patients with MD primarily displayed RPE changes in the macula (82.6%) and flecks in the posterior pole (73.9%; Figure 1B).

### 3.2. Molecular Findings

Genetic results that explain clinical diagnosis were identified for 48 (75%) patients. Pathogenic and likely causative mutations are indicated by the number of patients in this cohort affected by a given gene as shown in Figure 1B. Gene mutations were most frequently (23%) found in the gene *ABCA4* (Table 1) followed by *PRPH2* (*n* = 7, 11%; Figure 1B). Five patients from two families had *RP2* mutations (8%). Three patients (5%) with autosomal recessive RP (arRP) had *CNGB1*-related disease. The same *GUCA1A* mutation (Table 1) was found for three CRD patients (5%) from two different families. Mutations were identified for two individuals for each of the genes *ADGRV1*, *USH2A*, and *RPGR* (3% each). The remaining genes (*PDE6B*, *CNGB3*, *BEST1*, *HK1*, *NMNAT1*, *PRPF31*, *RHO*, *RP1*) were associated with a single patient (2%) with IRD (Figure 1B). Ten novel mutations and the previously reported as IRD-causing mutations are provided in Table 1. The unique mutations comprised five missense, one frameshift, three in-frame deletions or duplications, and one promoter mutation (Table 1). Altogether, we were able to provide a genetic diagnosis to 21 RP patients, seven patients with CRD, and 20 patients with MD (Figure 1C).

### 3.3. Genotype Phenotype Correlations

Pedigree analysis for an RP patient GP006 revealed that his mother (GP006a), maternal grandmother (GP006b), and maternal great uncle (not evaluated, Figure 2A) were also affected. GP006 and GP006a were diagnosed with RP at ages 10 and 15 years, respectively (Table 1). Fundus exam of GP006 showed optic disc pallor OU (oculus uterque or both eyes) arteriolar narrowing, and bone spicules in the peripheral retina (Figure 2B). Hypo-autofluorescence (AF) in the fovea and nasal periphery suggesting RPE atrophy (Figure 2C). SDOCT images revealed ellipsoid zone (EZ) changes and confirmed foveal RPE atrophy OU (Figure 2D). A novel mutation c.515dup (p.Ser172Argfs*2) in the gene *RP2* was identified on genetic testing for each of the affected patients (GP006, GP006a, and GP006b). Clinical evaluation for GP006a showed high myopia (Table 1), amblyopia OS (oculus sinister or left eye) and cataract OU. GP006a had typical RP features and RPE changes in the maculae (Figure 2E). Fundus autofluorescence (FAF) imaging displayed nasal hypo-AF regions, with dense hypo-AF areas distributed in the posterior pole (Figure 2F). Additionally, outer nuclear layer (ONL) loss, outer retinal atrophy, and mild vitreomacular adhesion was identified on SDOCT images (Figure 2G). On fundus examination, GP006b showed tilted optic nerves with PPA, arteriolar attenuation, as well as pigmentary and atrophic changes most noticeable in the temporal and inferior periphery (Figure 2H). There were hypo-AF lesions in the inferior periphery and AF striations in the posterior pole OU (Figure 2I). GP006b also displayed significant EZ irregularities including temporal atrophy OU (Figure 2J).

A separate family (patient GP063 and his mother GP063a) also harbored a mutation in *RP2* (Figure 2K). GP063 was diagnosed with RP at age 16 (Appendix A) but nyctalopia onset occurred at 4-years-of-age. Bilateral cataracts were removed at age 21. GP063 had typical RP features (Figure 2L) and central hypo-AF surrounded by parafoveal hyper-AF (Figure 2M) which was concomitant with photoreceptor loss in the maculae (Figure 2N). The patient’s mother (GP063a) reported nyctalopia at age 16 and was diagnosed with typical RP (Figure 2O) at 30 years-of-age. FAF revealed multifocal areas of hypo-AF corresponding to regions of bone spicule formation on color fundus photos (Figure 2P). Loss of the EZ was seen temporally OD (oculus dexter or right eye) whereas OS had almost complete loss of the photoreceptors (Figure 2Q). Genetic testing for this family revealed a novel *RP2* mutation c.413A>C (p.Glu138Ala). A pathogenic mutation has been reported previously at the same codon [87,88,89].

Another interesting case included patient GP037, who was diagnosed with isolate RP at age 30 with nyctalopia onset at 12-years-of-age. Her medical history included hypertension, diabetes, chronic renal insufficiency, gout, and gastric bypass. Fundus images were symmetrical OU showing chorioretinal atrophy, RPE changes in the macula and typical RP features (Figure 3A). FAF showed atrophic midperipheral hypo-AF, PPA, and a bullseye pattern of AF (Figure 3B). SDOCT imaging was notable for outer retinal atrophy with foveal preservation of the EZ (Figure 3C). Four generations of family history revealed no relatives with visual problems (pedigree not shown). Genetic testing reported a pathogenic mutation c.625C>T (p.Arg209*) and a variant of uncertain significance (VUS) c.712A>G (p.Lys238Glu) in the gene *CEP83*. The VUS Lys238Glu is a missense DNA change encoding a moderately conserved nucleotide present in 8 of 12 total species. ClinVar (2022-02-05, 1005895) classifies this variant with uncertain significance for Nephronophthisis. This information expands the phenotype of IRD for the *CEP83* gene which has only been reported in 2 other individuals [58,59] and is discussed below.

## 4. Discussion

Because IRDs are characterized by genetic and clinical heterogeneity, gene identification and mutation analysis are challenging but important. Continual advancements in sequencing technologies and genetic discoveries have had a substantial impact on the molecular diagnosis of IRD patients [5,6,47,90]. We therefore genetically tested 64 individuals, which identified pathogenic mutations in 18 genes, including ten novel genetic mutations (Table 1). Although patients with RP had the lowest gene detection rate, previous reports using multigene panels, next-generation sequencing, or exome sequencing have had similar findings (50% to 80%) [66,91,92,93,94,95].

A recent study by Sheck et al. [96] found that patients diagnosed with STGD1, MD, or CRD were solved for 91%, 63%, or 87% of patients, respectively. Although we included the STGD1 in the MD group, the diagnostic rates are similar to our results showing that patients with MD and CRD had a 83% and 87% chance of receiving a definitive genetic diagnosis (Figure 1C), respectively. The one patient (GP059) with MD whose test was considered negative, received a partial diagnosis due to a heterozygous pathogenic mutation in *ABCA4* c.1749G>C (p.Lys583Asn) [83]. She also harbored a pathogenic mutation in *CNGA1* c.1339dup (p.Thr447Asnfs*3). STGD1 MD is inherited in an autosomal recessive manner and no second potentially disease-causing mutation was detected in the *ABCA4* gene. It is well known that there are missing alleles in unsolved patients with one pathogenic allele in the *ABCA4* gene [97,98].There was a single report (ARVO abstract) suggesting that *ABCA4* may cooperate in a digenic fashion to cause IRD [99]. Although a possibility, it is more likely that a second *ABCA4* mutation was not detected (intronic or deletion/duplication) or that another yet undiscovered gene is responsible.

Earlier this year, a new gene, *CLEC3B,* was implicated as a cause of MD [100]. Patients with a *CLEC3B* mutation had drusen in the posterior pole, nyctalopia, and some decreased rod responses evaluated with full-field electroretinography (ffERG) [100]. This gene was not included in the gene panel for the four MD patients without a molecular diagnosis in our cohort. We therefore used Sanger sequencing to see if these patients had the same mutation. All four patients had the wild type *CLEC3B* allele (Appendix A). As more information about this gene becomes available, we will consider evaluating the entire coding region of *CLEC3B* in the unresolved MD cases.

In this research, genotype phenotype relationships were detected for two genes. First, mutations in the gene *RP2* were found for two unrelated families. Both *RP2* mutations reported here are localized to exon 2, which encodes a cofactor-C homologous domain (CFCHD), confirming a crucial role in RP2 function [101]. The mutation at Ser172 was previously reported as a severe pathogenic frameshift predicted to result in loss of function [36,37]. Jayasundera et al. [36] showed a male patient with this mutation who had significant choriocapillaris atrophy in the midperiphery and the posterior pole without pigment deposition, similar to patient GP006. A different exon 2 mutation (c.413A>C, p.Glu138Ala) in *RP2* was found for GP063 and GP063a which affects an evolutionary conserved residue (Glu138) that is a charged, hydrophilic amino acid. Substitution by an uncharged, hydrophobic residue (alanine) is likely to impair CFCHD function, leading to loss or reduced function of the RP2 protein [88]. Multiple lines of computational evidence support a deleterious effect on the gene or gene product (Damaging CADD score of 26.6, likely to interfere with function by Align GVGD Class C65, probably damaging by PolyPhen2, deleterious by SIFT and MutationTaster). This mutation is located in a mutational hot spot and critical functional domain. It is also absent from controls in Exome Sequencing Project, 1000 Genomes Project, and Exome Aggregation Consortium. A different mutation at the same nucleotide (c.413A>G, p.Glu138Gly) was found in two unrelated XLRP families with Italian ancestry [87,88]. Clinical characteristics associated with the p.Glu138Gly mutation included early-onset macular atrophy, progressive BCVA loss, night blindness, and visual field constriction since infancy. Additionally, the patient displayed high refractive myopia, posterior subcapsular cataract, and advanced RP associated with a choroideremia-like fundus [87] similar to other RP2-XLRP patients [36]. Two carrier females (GP006a and GP063a) manifested an RP phenotype, exhibiting atrophic macular changes and poor visual acuity. The carrier female GP006a demonstrated anisometropia (asymmetric refraction) of approximately 2.75 D OS, and her son GP006 was moderately myopic (−3.75 D OU). These refractive errors are in agreement with the myopia association in *RP2* retinal disease [36,83,102]. Conversely, GP063 and GP063a were mildly hyperopic in one eye (OD: +0.50 and +0.75, respectively; Table 1) although GP063a had cataract surgery and reported that she was myopic prior to the surgery. There have been no other carrier females with similar mutations described in the RP2-XLRP literature.

To date, RetNet catalogues 340 genes that cause IRDs. However, the gene *CEP83* is not included in that database likely owing to its novelty for association with IRD. The *CEP83* gene encodes a protein that is essential for ciliogenesis initiation. The CEP83 protein is a core component of the centriole that is found in almost all cell types. Ciliopathies manifest particularly during early childhood or adolescence and can affect almost every organ system [103,104]. Mutations resulting in ciliopathies are often associated with retinal degeneration, cystic kidney disease, fibrocystic liver disease, diabetes mellitus, obesity, skeletal dysplasia, and various abnormalities of the central nervous system [104,105]. GP037 had *CEP83* mutations c.625C>T (p.Arg209*) and c.712A>G (p.Lys238Glu). Although CEP83 is primarily associated with nephronophthisis, there have been two publications showing four patients who had autosomal recessive RP (arRP) without nephronophthisis. One of these patients was homozygous for the p.Arg209* mutation with the same clinical features as the patient (GP037) reported here. The biallelic mutations in *CEP83* was c.712A>G (p.Lys238Glu). Although pathogenicity has not been proven, we consider the p.Lys238Glu mutation to be disease causing. First, based on the ACMG guidelines, pathogenicity of this mutation is supported by lines of evidence supporting evolutionary conservation (pathogenic supporting argument 3). There is an almost complete absence of the mutation in population databases gnomAD 0.00040084% (1 person) and 0 individuals in ESP, thereby fulfilling a pathogenic moderate argument 2 [106]. Pathogenicity is also supported by in silico modeling (CADD sore 24.1, Align GVGD Class C15, SIFT Deleterious). Moreover, the phenotype suggests pathogenicity (supporting argument 4) of the mutation, as retinal dystrophy has also been reported in another patient with arRP and biallelic mutations in *CEP83* (p.Arg209*/p.Arg511Pro) [58] located in the same coiled-coil domain as the p.Lys238Glu mutation harbored by GP057. She received genetic testing in 2017 that revealed a single ABCA4 mutation and a single FAM161A mutation. This was prior to the addition of *CEP83* to ophthalmology gene panels. Retesting in 2020 found the two CEP83 mutations in addition to the previously determined pathogenic mutations. Therefore, when a result is incomplete or inconclusive, the patient should be retested after some time because new genes, mutations, and technologies advance rapidly.

These results serve as a basis for future investigation toward understanding genotype phenotype associations and molecular mechanisms underpinning the pathology that leads to vision loss for patients with IRD. Moreover, the knowledge gained from this work contributes to developing treatment for IRD.

## Figures and Tables

**Figure 1 genes-14-00074-f001:**
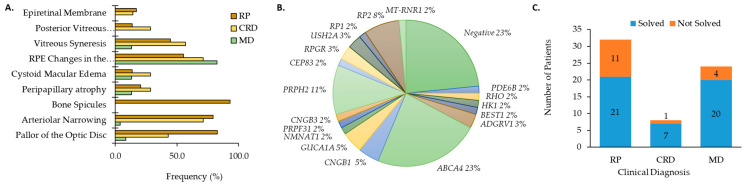
Clinical characteristics for patients with (**A**) cone rod dystrophy (CRD) and retinitis pigmentosa (RP) and macular dystrophy (MD). (**B**) Distribution of genes causative for ocular disease by number of patients (%). (**C**) The number of genetically solved cases subset by clinical diagnosis.

**Figure 2 genes-14-00074-f002:**
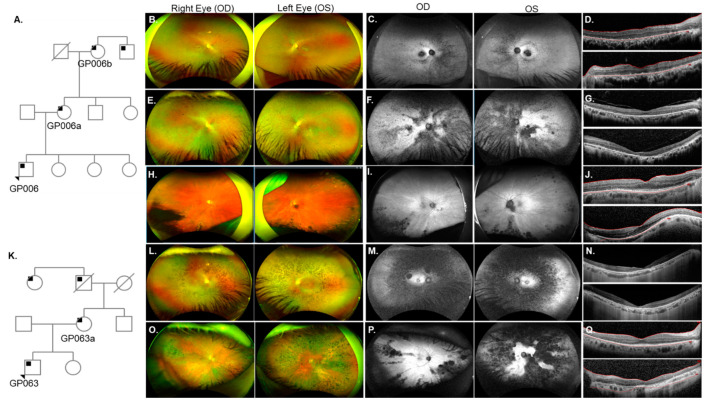
Phenotype of two families with X-linked RP (XLRP) due to mutations in *RP2*. (**A**) Pedigree analysis for patient GP006 showed three generations of RP (black squares). (**B**) Fundus images for GP006 showed optic disc pallor OU, arteriolar narrowing, and bone spicules in the peripheral retina. (**C**) Fundus autofluorescence (FAF) revealed hypo-AF in the nasal periphery and fovea. (**D**) SDOCT showed ellipsoid zone (EZ) changes and retinal pigment epithelium (RPE) atrophy OU. GP006a (**E**–**G**) had typical RP features (**E**) with hypo-AF regions in the nasal retina and throughout the posterior pole (**F**) and thinning of the outer retina (**G**). GP006b (**H**–**J**) had pallor of the optic disc, chorioretinal atrophy of the far periphery and pigment migration in the inferior retina (**H**). Radial lines and inferior hypo-AF were noted on FAF (**I**). SDOCT (**J**) showed outer retinal atrophy in the temporal macula OU for GP006b. (**K**) Pedigree analysis for patient GP063 indicated maternally inherited RP with affected family members from three generations. GP063 (**L**–**N**) had typical RP features, an atrophic macula, and a loss of EZ band on SDOCT. GP063a (**O**–**Q**) had typical RP, multifocal areas of hypo-AF, a loss of EZ temporally OD (upper panel), and complete loss OS (lower panel).

**Figure 3 genes-14-00074-f003:**
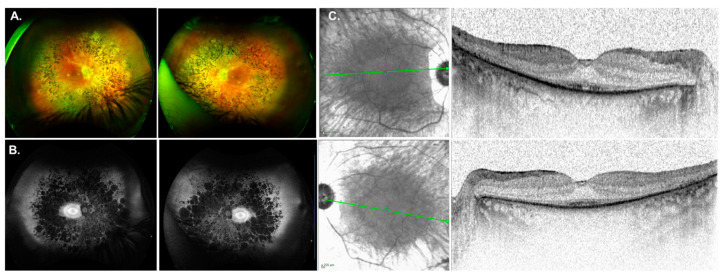
Phenotype associated with *CEP83* mutations. (**A**) Fundus images for GP037 were showing chorioretinal atrophy, RPE changes in the macula and typical RP features. (**B**) Fundus autofluorescence imaging showed an atrophic midperiphery and a bullseye pattern of autofluorescence. (**C**) SDOCT imaging was notable for outer retinal atrophy with foveal preservation of the ellipsoid zone (EZ) line.

**Table 1 genes-14-00074-t001:** Patient Genetics.

Subject ID	Ethnicity (Gender)	Refraction OD; OS	BCVA OD; OS	LogMAR OD; OS	Age at Dx/GT	Dx	Gene 1	Gene 2	Variant
DM001	WC (M)	ND	20/320; CF	1.2;	3/56	adCRD	*GUCA1A*	NA	c.428delinsACAC (p.Ile143delinsAsnThr) [18,19]
DM002	WC (M)	+2.50 + 1.00 × 090; +1.75 + 2.00 × 090	20/50; 20/63	0.4; 0.5	5/8	arRP	*PDE6B*	*PDE6B*	c.892C>T (p.Gln298*) [20,21,22]/c.1954C>T Gln652*) [23]
DM003	WC (F)	−5.50 + 1.00 × 090; −5.00 + 1.25 × 090	20/100; 20/100	0.7; 0.7	13/16	arMD	*ABCA4*	*ABCA4*	c.2588G>C (p.Gly863Ala) ^a^/c.1622T>C (p.Leu541Pro) ^b^ & c.3113C>T (p.Ala1038Val) ^c^
DM004	WC (M)	ND	20/400; 20/125	1.3; 0.8	23/28/	arCRD	*NMNAT1*	** *NMNAT1* **	c.769G>A (p.Glu257Lys) [24,25,26]/**c.-71G>C**
DM005	NA (M)	−1.75 + 1.75 × 040; −1.50 + 0.75 × 2.25	20/25; 20/32	0.1; 0.2	35/50	adMD	*PRPH2*	NA	c.828+3A>T (p.?) [3,27,28,29,30,31,32]
GP001	WC (F)	ND	20/20; 20/20	0; 0	33/37	MDdiso	negative	negative	
GP002	BA, NA (F)	ND	20/50; 20/40	0.4; 0.3	49/51	arRP	*CNGB1*	*CNGB1*	c.583+2T>C [33,34]/2305-34G>A [33,34]
GP004	WC (F)	−0.25 + 4.50 × 100; +1.25 + 1.00 × 077	20/32; 20/20	0.2; 0	35/36	adMD	*PRPH2*	NA	c.828+3A>T (p.?) [3,27,28,29,30,31,32]
GP005	WC (M)	ND	20/50; 20/160	0.4; 0.9	25/46	arMD	*ABCA4*	*ABCA4*	c.5882G>A (p.Gly1961Glu) ^a^/c.3259G>A (p.Glu1087Lys) ^d^ & c.2042G>A (p.Arg681Gln) [35]
GP005a	WC (F)	ND	20/800; 20/640	1.6; 1.5	10/50	arMD	*ABCA4*	*ABCA4*	c.5882G>A (p.Gly1961Glu) ^a^/c.3259G>A (p.Glu1087Lys) ^d^ & c.2042G>A (p.Arg681Gln) [35]
GP006	WC, NA (M)	−3.75 + 1.75 × 100; −3.75 + 1.00 × 096	20/160; 20/125	0.9; 0.8	15/21	XLRP	*RP2*	NA	c.515dup(p.Ser172Argfs*2) [36,37]
GP006a	WC, NA (F)	+0.25 − 1.25 × 045; −2.75 − 3.75 × 155	20/32; 20/500	0.2; 1.4	10/41	XLRP	*RP2*	NA	c.515dup(p.Ser172Argfs*2) [36,37]
GP006b	NA (F)	0.00 + 3.50 × 105; −3.00 + 4.75 × 080	20/32; 20/30	0.2; 0.2	35/64	XLRP	*RP2*	NA	c.515dup(p.Ser172Argfs*2) [36,37]
GP008	WC (F)	ND	20/50; 20/50	0.4; 0.4	25/51	arRP	*ADGRV1*	** *ADGRV1* **	c.17668_17669del (p.Met5890Valfs*10) ^l^/**c.4378G>A (p.Gly1460Ser)**
GP008a	WC (F)	ND	20/50; 20/50	0.4; 0.4	30/55	arRP	*ADGRV1*	** *ADGRV1* **	c.17668_17669del (p.Met5890Valfs*10) ^l^/**c.4378G>A (p.Gly1460Ser)**
GP009	WC (M)	+0.75 + 0.50 × 170; +1.25 + 0.25 × 165	20/320; 20/200	1.2; 1	39/53	adRP	negative	negative	
GP012	Asian (M)	ND	20/250; 20/640	1.1; 1.5	31/42	adCRD	*PRPH2*	NA	c.367C>T (Arg123Trp) [38]
GP013	WC (F)	ND	20/20; 20/20	0; 0	35/36	adMD	*PRPH2*	NA	c.828+3A>T (p.?) [3,27,28,29,30,31,32]
GP013a	WC, BA (F)	−0.50 + 0.25 × 075; Plano	20/15; 20/20	−0.12; 0	20/21	adMD	*PRPH2*	NA	c.828+3A>T (p.?) [3,27,28,29,30,31,32]
GP014	WC (M)	−8.25 + 1.00 × 130; −9.75 + 1.25 × 055	20/640; 20/200	1.5; 1	60/71	arCRD	*CNGB3*	*CNGB3*	c.1148del (p.Thr383Ilefs*13) [39,40,41,42,43,44]/c.(852+1_853-1)_(903+1_904-1)dup [45]
GP015	WC (F)	ND	20/100; 20/200	0.7; 1	40/79	Rpiso	negative	negative	
GP016	WC (M)	+0.50 + 1.00 × 080; 0.00 + 1.00 × 095	20/63; 20/80	0.5; 0.6	7/25	XLRP	negative	negative	
GP017	NA (M)	+3.00 − 1.00 × 124; +3.50 − 0.75 × 076	20/200; 20/200	1; 1	20/48	MDiso	*BEST1*	** *BEST1* **	c.286C>G(p.Gln96Glu) [46]/**c.579_580insCATT** (p.Lys194Hisfs*2)
GP018	NA (M)	ND	20/200; 20/400	1; 1.3	20/56	Rpiso	negative	negative	
GP019	UK (F)	+1.00 + 0.00 × 000; Plano	20/320; 20/32	1.2; 0.2	69/70	adMD	*PRPH2*	NA	c.828+3A>T (p.?) [3,27,28,29,30,31,32]
GP020	WC (M)	−0.25 + 0.75 × 110; −0.25 + 0.50 × 050	20/25; 20/25	0.1; 0.1	22/23	adRP	*RHO*	NA	c.68C>A (p.Pro23His) ^m^
GP021	NA (M)	−7.50 + 2.75 × 080; −7.25 + 2.50 × 120	20/400; 20/320	1.3; 1.2	5/68	arCRD	*ABCA4*	** *ABCA4* **	c.5381C>A (p.Ala1794Asp) ^e^/**c.5909T>G (p.Leu1970Arg)**
GP022	NA (M)	0.00 + 0.75 × 135; +0.25 + 0.00 × 000	20/25; 20/63	0.1; 0.5	48/49	adRP	*ABCA4*	*ABCA4*	c.2701A>G (p.Thr901Ala) [47]/c.5603A>T (p.Asn1868Ile) ^f^
GP022a	NA (F)	−6.00 + 0.50 × 090; −6.25 + 1.00 × 105	20/20; 20/25	0; 0.1	27/36	adRP	negative	negative	
GP026	WC (F)	−1.75 + 0.75 × 120; −3.00 + 1.25 × 085	20/125; 20/160	0.8; 0.9	41/41	arMD	*ABCA4*	** *ABCA4* **	c.6079C>T (p.Leu2027Phe) ^g^/**c.5281_5289del (p.Pro1763del)**
GP028	WC (F)	ND	HM; 20/125	0.8	25/84	Rpiso	negative	negative	
GP029	WC (M)	ND	LP; LP		80/81	XLRP	*RPGR*	NA	c.2442_2445del (p.Gly817Lysfs*2) [37,48,49]
GP029a	WC (M)	ND	ND	ND	unknown	XLRP	*RPGR*	NA	c.2442_2445del (p.Gly817Lysfs*2) [37,48,49]
GP031	WC (F)	−2.75 + 00 × 000; −3.00 + 0.00 × 000	20/60; 20/160	0.48; 0.9	30/70	arMD	*ABCA4*	*ABCA4*	c.71G>A (p.Arg24His) [50,51,52,53,54]/c.4469G>A (p.Cys1490Tyr) ^h^
GP032	WC (M)	+0.25 + 0.75 × 010; −0.75 + 0.75 × 145	20/20; 20/20	0; 0	37/51	arRP	*USH2A*	*USH2A*	c.10073G>A (p.Cys3358Tyr) [55,56,57]/c.10342G>A (p.Glu3448Lys) ^n^
GP034	WC (M)	0.00 + 0.75 × 025; NLP	20/25; NLP	0.1;	35/59	arRP	negative	negative	
GP035	NA (F)	−5.25 + 0.00 × 000; −5.75 + 2.00 × 090	20/40; 20/25	0.3; 0.1	39/41	adMD	** *PRPH2* **	NA	**c.829-A_1041+?del**
GP036	NA (F)	−2.25 + 1.50 × 126; −3.25 + 0.75 × 014	CF; CF		20/70	adRP	negative	negative	
GP037	WC (F)	+1.00 + 1.00 × 146; +0.50 + 1.25 × 012	20/63; 20/63	0.5; 0.5	30/55	Rpiso	*CEP83*	** *CEP83* **	c.625C>T (p.Arg209*) [58,59]/**c.712A>G (p.Lys238Glu)**
GP039	WC (F)	−0.25 + 0.75 × 110; 0.00 + 0.75 × 070	20/32; 20/32	0.2; 0.2	27/40	Rpiso	negative	negative	
GP044	WC (M)	−2.50 + 0.50 × 085; −1.00 + 1.00 × 082	20/50; 20/80	0.4; 0.6	40/65	adRP	** *RP1* **	NA	**c.2321_2322ins? (p.Leu774fs)**
GP045	WC (F)	ND	20/20; 20/20	0; 0	34/53	arRP	*CNGB1*	*CNGB1*	c.2492+1G>A (p. ?) [33,34]; c.2092T>C (p.Cys698Arg) [33,34]
GP045a	WC (F)	−1.50 + 1.50 × 065; −1.25 + 1.00 × 090	20/40; 20/40	0.3; 0.3	36/42	arRP	*CNGB1*	*CNGB1*	c.2492+1G>A (p. ?) [33,34]; c.2092T>C (p.Cys698Arg) [33,34]
GP046	WC (F)	+4.00 + 0.75 × 015; +1.25 + 0.75 × 117	20/32; 20/63	0.2; 0.5	23/24	MDiso	*ABCA4*	*ABCA4*	c.5882G>A (p.Gly1961Glu) ^a^/c.161G>A (p.Cys54Tyr) [50,60,61,62,63,64]
GP047	WC (F)	−4.00 − 2.25 × 010; −2.50 − 2.25 × 170	20/40; 20/40	0.3; 0.3	60/67	arMD	*ABCA4*	*ABCA4*	c.161G>A (p.Cys54Tyr) [50,60,61,62,63,64]/c.6089G>A (p.Arg2003Gln) ^i^
GP047a	WC (M)	−2.50 + 1.75 × 112; −2.00 + 1.50 × 065	20/40; 20/50	0.3; 0.4	14/48	arMD	*ABCA4*	** *ABCA4* **	c.161G>A (p.Cys54Tyr) [50,60,61,62,63,64]/**c.1304G>T (p.Gly435Val)** & c.5603A>T (p.Asn1868Ile) ^f^
GP048	BA (M)	−2.00 + 0.00 × 000; −2.50 + 0.50 × 150	20/20; 20/20	0; 0	31/49	MDiso	negative	negative	
GP049	NA (M)	−4.25 + 0.00 × 000; −4.25 + 0.50 × 175	20/50; 20/32	0.4; 0.2	40/49	arRP	*USH2A*	*USH2A*	c.2299del (p.Glu767Serfs*21) ^k^/c.4106C>T (p.Ser1369Leu) [65,66,67]
GP050	WC (M)	+2.75 + 0.75 × 150; +2.50 + 1.00 × 011	20/32; 20/25	0.2; 0.1	47/49	arMD	*ABCA4*	*ABCA4*	c.655A>T (p.Arg219*) [54,68,69]/c.5603A>T (p.Asn1868Ile) ^f^
GP052	WC (F)	−2.50 + 1.00 × 090; −2.50 + 1.25 × 090	20/160; 20/160	0.9; 0.9	13/26	arMD	*ABCA4*	*ABCA4*	c.4773+3A>T (p.?) [70,71,72]/c.4139C>T (p.Pro1380Leu) ^j^
GP053	WC (M)	−1.75 + 0.75 × 090; −2.25 + 1.00 × 090	20/15; 20/15	−0.1; −0.1	31/34	arMD	*ABCA4*	*ABCA4*	c.1726G>C (p.Asp576His) [54,73,74]/c.4577C>T (p.Thr1526Met) [50,75]
GP054	WC (F)	−1.00 + 0.00 × 000; −2.00 + 0.00 × 000	20/25; 20/25	0.1; 0.1	29/32	adRP	*HK1*	NA	c.2539G>A, (p.Glu847Lys) [76,77,78]
GP058	WC (F)	−0.25 + 1.75 × 180; −0.50 + 1.00 × 020	20/125; 20/40	0.8; 0.3	24/70	adRP	*PRPF31*	NA	c.(?_−396)_(*1_?)del [28,79,80,81,82]
GP059	BA (F)	−0.50 + 0.50 × 020; −0.75 + 0.50 × 165	20/20; 20/20	0; 0	46/47	MDiso	*ABCA4*	*CNGA1*	c.1749G>C (p.Lys583Asn) [8,83]; **c.1339dup (p.Thr447Asnfs*3)**
GP060	WC (M)	−0.50 + 0.50 × 020; 0.00 + 0.50 × 150	20/80; 20/63	0.6; 0.5	53/67	arMD	*ABCA4*	*ABCA4*	c.5882G>A (p.Gly1961Glu) ^a^/c.4577C>T (p.Thr1526Met) [50,75]
GP061	WC (F)	ND	20/200; 20/160	1; 0.9	21/28	Rpiso	negative	negative	
GP062	WC (F)	unknown	20/20; 20/20	0; 0	33/44	MDiso	negative	negative	
GP063	WC (M)	+0.50 + 1.25 × 120; 0.00 + 1.25 × 060	20/320; 20/200	1.2; 1	16/35	XLRP	** *RP2* **	NA	**c.413A>C, p.Glu138Ala**
GP063a	WC (F)	+0.75 + 0.00 × 000; Plano	20/20; 20/40	0; 0.3	30/60	XLRP	** *RP2* **	NA	**c.413A>C, p.Glu138Ala**
GP064	WC (F)	+3.25 + 0.00 × 000; +3.00 + 1.00 × 160	20/20; 20/20	0; 0	35/39	MDiso	negative	negative	
GP065	WC (M)	−1.75 + 1.50 × 155; −2.00 + 1.50 × 180	20/100; 20/125	0.7; 0.8	51/57	CRDiso	negative	negative	
GP066	BA (F)	−3.25 + 0.75 × 175; −4.75 + 0.75 × 025	20/40; 20/15	0.3; −0.1	31/46	arRP	*MT-RNR1*	negative	m.1555A>G (homoplasmic) [84,85,86]
GP067	WC (F)	−1.75 + 0.50 × 174; −3.00 + 0.50 × 090	20/200; 20/200	1; 1	50/62	adCRD	*GUCA1A*	NA	c.428delinsACAC (p.Ile143delinsAsnThr) [18,19]
GP067a	WC (M)	−5.50 + 1.50 × 100; −4.00 + 1.00 × 171	20/40; 20/25	0.3; 0.1	38/35	adCRD	*GUCA1A*	NA	c.428delinsACAC (p.Ile143delinsAsnThr) [18,19]

M, Male; F, Female; Dx, Diagnosis; GT, Genetic Testing; WC, White/Caucasion; BA, Black/African American; NA, Native American; UK, Unknown/undisclosed; autosomal dominant or recessive (ad or ar), X-linked (XL), isolate (iso), MD, Macular Dystrophy; RP, Retinitis Pigmentosa; CRD, Cone-Rod Dystrophy; ND, not done; OD oculus dexter (right eye); OS oculus sinister (left eye); CF, counting fingers; HM, hand motion; LP, light perception. See ClinVar Accession numbers: ^a^ VCV000007879, ^b^ VCV000099067, ^c^ VCV000007894, ^d^ VCV000099211, ^e^ VCV000099371, ^f^ VCV000099390, ^g^ VCV000099428, ^h^ VCV000099288, ^i^ VCV000099428, ^j^ VCV000007904, ^k^ VCV000002351, ^l^ VCV00050362, ^m^ VCV000013013, ^n^ VCV000209203. Novel variants are in bold.

## Data Availability

NGS panel statistics available upon request.

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
