# Peer review of "Genetic Diagnosis for 64 Patients with Inherited Retinal Disease"

_genes, 2022, doi:10.3390/genes14010074_

Round 1

Reviewer 1 Report

In this paper, Lynn et al. performed mutation analysis on 63 patients with IRDs. NGS coupled with Sanger sequencing identified mutations in > 70 % of the patients. This study also identified ten novel mutations in IRD patients. The overall study design, techniques, and the other sections of the manuscript, including figures and tables, are well organized and illustrate the facts discussed in the text. The interpretations are consistent with the findings. However, I have some questions related to the experiments conducted in this paper. Below are my concerns:

1.    The Method section looks incomplete, especially the genetic testing part. The authors should provide more details about how the NGS was performed. If it is targeted sequencing, I recommend including the lists of all genes used for targeted NGS and the methods used for NGS. The authors should also explain if the samples from the family members were included in NGS. The method section is very brief and does not provide many details.

2.    Lines 29-30 and 113: Both lines are confusing; is it 72 % or 76 %?

3.    How was the RT-PCR method used to confirm deletions? There are no details provided for this part.

4.    The authors should list all the primer sequences used for PCR and Sanger sequencing as a supplementary table.

5.    Line 113- ‘Genetic results that explain clinical diagnosis were identified for 48 (76%) patients.

Lines 125-126- ‘Altogether, we were able to provide a genetic diagnosis to 20 RP patients, seven patients with CRD, and 20 patients with MD’..= 47 patients. 

From both these lines, it is unclear if 47 or 48 patients were positive for mutations.

6.    For all the novel mutations identified, the authors should provide chromatograms from Sanger sequencing, family details, and segregation analysis (if sequencing was performed). 

7.    In the case of novel missense mutations, the authors are advised to perform bioinformatic analysis to test the pathogenicity of these mutations. Also, did the authors check if these mutations are absent in the control population?

Finally, A more tightly worded manuscript with the above-suggested corrections will make the paper a good read.

Reviewer 2 Report

It is a good work that the authors observed 63 patients by genetically diagnosed with inherited retinal disease. The correlations between genotype and phenotype are always paid attention in clinic. It is better that the authors display the genotypes and phenotypes together in one table. Thus, the readers are able to understand the correlation between genotypes and phenotypes clearly.
